# Continuous Dynamics Monitoring of Multi-Lake Water Extent Using a Spatial and Temporal Adaptive Fusion Method Based on Two Sets of MODIS Products

**DOI:** 10.3390/s19224873

**Published:** 2019-11-08

**Authors:** Pinzeng Rao, Linjiang Lou, Weiguo Jiang, Yicheng Wang, Xiaoya Wang, Xiayu Cao

**Affiliations:** 1State Key Laboratory of Simulation and Regulation of Water Cycle in River Basin, China Institute of Water Resources and Hydropower Research, Beijing 100038, China; raopinzeng@mail.bnu.edu.cn (P.R.); wangych@iwhr.com (Y.W.); 2Land Satellite Remote Sensing Application Center, Ministry of Natural Resources of China, Beijing 100048, China; 3State Key Laboratory of Remote Sensing Science, Faculty of Geographical Science, Beijing Normal University, Beijing 100875, China; wangxiaoya@mail.bnu.edu.cn; 4Beijing Key Laboratory for Remote Sensing of Environment and Digital Cities, Faculty of Geographical Science, Beijing Normal University, Beijing 100875, China; 5Changjiang Institute of Survey, Planning, Design and Research, Wuhan 430010, China; caoxiayuyu@163.com

**Keywords:** lake water extent, continuous dynamics monitoring, data fusion, MODIS, HDSTAFM

## Abstract

Due to the widespread presence of noise, such as clouds and cloud shadows, continuous, high spatiotemporal-resolution dynamic monitoring of lake water extents is still limited using remote sensing data. This study aims to take an approach to mapping continuous time series of highly-accurate lake water extents. Four lakes from diverse regions of China were selected as cases. In order to reduce the impact of noise and ensure high spatial and temporal resolution of the final results, two sets of MODIS products (including MOD09A1 and MOD13Q1) are used to extract water bodies. This approach mainly comprises preliminary classification, post processing and data fusion. The preliminary classification used the Random Forest (RF) classifier to efficiently and automatically obtain the initial classification results. Post-processing is implemented to repair the classification results affected by noise as much as possible. The processed results of the two sets of products are fused by using the Homologous Data-Based Spatial and Temporal Adaptive Fusion Method (HDSTAFM), which reduces the effect of noise and also improve the temporal and spatial resolution for the final water results. We determined the accuracy using Landsat-based water results, and the values of overall accuracy (OA), user’s accuracy (UA), producer’s accuracy (PA), and kappa coefficients (KC) are mostly greater than 0.9. Good correlation was achieved for a time series of water area and altimetry data, obtained by multiple satellites, and also for water-level data selected from hydrological stations.

## 1. Introduction

Lakes are closely related to human life and the natural environment [1]. Accurate recognition of long-term or dynamic changes in lakes is essential [2]. Remote sensing satellites can provide a significant amount of data which is necessary for monitoring. Long-term dynamics of terrestrial water bodies have been mapped [3,4,5]. However, continuous short-term monitoring of terrestrial lake-water with high spatial resolution is still a difficult task. Appropriate data sources and robust methods are both the key to long-term monitoring of water bodies.

MODIS data is a good source, due to its high temporal and medium spatial resolution [6,7,8,9,10]. Ogilvie et al. [10] monitored the spatiotemporal dynamics of the annual flood across the Niger Inner Delta by using 526 MOD09A1 images during 2000–2011. Feng et al. [11] used MODIS Level-0 time series data to map the inundated extents of Poyang Lake from 2000 to 2010. However, dense optical remote sensing data is susceptible to noise, and these noises can cause very large errors for the final results. Some studies have focused on the removal of noise [12,13,14,15]. Pekel et al. [3] considered terrain shadows, glaciers, lava, building cast shadows, and cloud shadows for Landsat-based water results. In order to obtain continuous surface water, it is critical to repair the water covered by noise, such as clouds and cloud shadows. Khandelwal et al. [16] introduced an effective post-processing technique based on MODIS products to extract the long-term water extent of 94 reservoirs worldwide. These jobs are useful, but the results are still greatly affected by noise.

A suitable classifier is very useful for long-term water extraction. The water extraction method is relatively mature and can be roughly divided into threshold methods using some index, supervised and unsupervised classification, and some other methods [17]. Index methods (such as NDWI) are simple and feasible, but it is difficult for time series water bodies extraction, because the threshold values need to be constantly adjusted in different seasons [18]. Supervised classification methods are feasible, and which one does not need to determine the threshold, only need to provide some reliable sample points. Some studies have demonstrated that the Random Forest (RF) classifier is robust and has efficient classification capabilities in land cover and surface water [19,20,21]. Compared to a single data source, the joint use of efficient multiple data sources help to avoid noise and achieve better results [22,23]. Some data assimilation and fusion methods for multiple data sources are well presented [24], such as the Spatial and Temporal Adaptive Reflectance Fusion Model (STARFM) [25] and the Spatial and Temporal Reflectance Unmixing Model (STRUM) [26]. However, data fusion for the acquisition of lake “water” bodies’ time series is a challenge, due to its dramatic dynamic characteristics. The fusion of different data sources has significant differences, because of the sensor itself and the zenith angle [27]. Moreover, the use of fused products may cause greater errors due to error overlapping. In this study, a Homologous Data-based Spatial and Temporal Adaptive Fusion Method (HDSTAFM) has been proposed to better avoid these problems. The HDSTAFM mainly consider the combined use of multiple sets of MODIS products. It not only improves the spatiotemporal resolution, but also reduces the effects of noise on the final water results.

Four different types of lakes around China, including Bosten Lake, Namco Lake, Hulun Lake, and Poyang Lake, are selected as cases. In order to reduce the impact of noise on the results, we first use a robust classifier to classify water and non-water bodies, and perform some post-processing (including de-noise and water body restoration) on the results. Then, the HDSTAFM is used to integrate the two results from MOD09A1 and MOD13Q1 which correspond to a period of 8 and 16 days with a resolution equal to 500 and 250 m, respectively. Finally, data related to water measurements for eight days, characterized by a resolution equal to 250 m, were produced.

## 2. Case Study Areas and Materials

### 2.1. Case Study Areas

Four case study areas around China, characterized by different water types, climate features, and underlying landscapes, were chosen to examine the robustness and the applicability of the approach. Figure 1 shows their locations and remote sensing images, and Table 1 provides basic information about case study areas.

### 2.2. Materials

The data used in this paper includes several MODIS datasets, DEM data, and validation data (including some Landsat images, the water level data from the multiple satellite and the hydrological station).

#### 2.2.1. MODIS Datasets

MOD09A1 is a set of surface reflectance products, containing bands 1–7 with an approximate resolution of 500 m in an eight-day gridded product. Each pixel is related to the best observation during eight days based on high observation coverage, low view angle, absence of clouds or cloud shadows, and aerosol loading. It also provides quality assignments (QA) data that contain MOD35 cloud/snow/ice flag, and the day of the year (DOY) for all pixels.

MOD13Q1 is mainly designed to monitor vegetation. It contains NDVI and EVI, as well as red, near-infrared, blue, and mid-infrared bands, corresponding to bands 1, 2, 3, and 7, respectively. The product is available every 16 days at approximately 250 m spatial resolution. Additionally, it also provides QA and DOY related data.

MOD44W product is the global surface water dataset with a spatial resolution of 250 m [27]. The product provides relatively reliable surface water extent and it can be used for training classification models [16,28]. Of note, These MODIS products were downloaded from the Level 1 and Atmosphere Archive and Distribution (LAADS, https://ladsweb.modaps.eosdis.nasa.gov).

#### 2.2.2. ASTER GDEM Data

The Advanced Spaceborne Thermal Emission and Reflection Radiometer (ASTER) Global Digital Elevation Model (GDEM) is probably the most accurate DEM dataset which can be obtained for free on a global scale. ASTER GDEM dataset with 30 m resolution, has been proven to have high vertical and horizontal accuracy, downloaded from https://search.earthdata.nasa.gov.

#### 2.2.3. Validation Data

Some validation data were used as the key to verify the accuracy of the results. As higher-resolution remote sensing images, some cloudless Landsat images in the same period can be used to verify the accuracy. In addition, the satellite altimetry water level data were also collected (Table 2), including: (1) U.S. Department of Agriculture (USDA) Global Reservoir and Lake Monitoring (GREALM, http://www.pecad.fas.usda.gov/cropexplorer/global_reservoir); (2) the Database for Hydrological Time Series of Inland Waters (DAHITI, http://dahiti.dgfi.tum.de/en/); and (3) the Laboratoire d’Etudes en Géophysique et Océanographies Spatiales (LEGOS, http://hydroweb.theia-land.fr). Some of them which have the same observation date as the MODIS date were selected for the reliability analysis of the classification result.

In addition, there are some hydrological stations to monitor the water level in Poyang Lake. In order to better verify the time-series results, we also collected the observed water level data from the Hukou hydrological station that is the most representative for the lake.

## 3. Methods

The flowchart of the proposed research algorithm, mainly includes three major parts: Classification using the RF classifier, post-processing, and results fusion using the HDSTAFM (Figure 2).

### 3.1. Water Preliminary Classification

The RF classifier belongs to the ensemble learning algorithms. Compared with a single tree classifier, it is more robust, and it has good generalization ability in classification, due to the characteristics of multiple trees and repetitive sampling [19,20]. The main advantages of RF are the following: (a) Robust behavior in handling outliers and noise; (b) efficient and fast classification performance; (c) high level of immunity to overfitting and efficiency in classification of time-series images [29,30].

NDVI, NDWI, and MNDWI were developed by using bands 1–7, and they made up 10 feature variables for the classification of the MOD09A1 product. Regarding MOD13Q1, its own two indices and four bands were considered as feature variables.

To choose accurate samples, we created a buffer using the MOD44W product of each lake, including core water, transition, and core land zones (CLZ). The core water zone (CWZ) was created by selecting the minimum value of 20 pixels × 20 pixels. The transition zone (TZ) is the zone selecting the maximum value of 20 pixels × 20 pixels but removing the CWZ. CLZ is the zone selecting the maximum value of 60 pixels × 60 pixels but removing CWZ and TZ. For different study areas, we randomly selected 2000 water sample points in CWZ and 2000 non-water sample points in CLZ (Figure 3c).

According to the provided multi-feature variables and sample points, the water and land can be classified using the RF classifier. 

### 3.2. Post-Processing

Post-processing includes de-noise and water restoration (Figure 4). MODIS QA data is used to remove the noise for the preliminary water results, including ice, snow, clouds, and cloud shadows. In addition, some lakes in the mountain or plateau need to remove terrain shadows by slope data. In this study, we considered that no surface water is present in places where the slope is >5°.

Water restoration is firstly restored by using the previous and subsequent results (Figure 4). Notably, we consider that these kinds of weather events, such as clouds and cloud shadows, increase the water of the lake, and the water area is generally relatively large. Hence, when the pixels of both two adjacent results correspond to water, the relative pixel of the middle result (located in the same position) that is covered by noise, is also water. Moreover, for each lake, we find the mode of the elevation values for all water pixels in the CWZ after de-noise. Then we convert non-water pixels whose elevation values are less than the mode, into water pixels. If there are multiple modes, we choose the minimum of them as the final value.

### 3.3. The Homologous Data-Based Spatial and Temporal Adaptive Fusion Method

The homologous data-based spatial and temporal adaptive fusion method mainly uses the composite day of the annual data of the MOD13Q1 products.

1. The principle of the HDSTAFM

The examples are selected from the MOD09A1, between the 20xx001th and the 20xx009th days (0 ≤ xx ≤ 16), and from the MOD13Q1 in the 20xx001th day (Figure 5). The pixels of the 20xx001th MOD09A1 product are all optimal selected from the 20xx001–20xx008th day, and the pixels of the 20xx009th MOD09A1 product correspond to the 20xx009–20xx016th day. Moreover, the pixels of the 20xx001th MOD013Q1 product are also best collected from the 20xx001–20xx016th day. This fusion process can be explained as follows:

First of all, the 20xx001th and the 20xx009th MOD09A1 results data were directly split into 250 m, so one pixel was split into four pixels. The pixel values (0 or 1) were not changed; that is, if the initial pixel value was 1, the values of the four pixels (obtained by the division) were also equal to 1. Their number of pixels after splitting was equal to the number of 20xx001th MOD13Q1. We took some corresponding pixels as the analysis objects (Figure 5). *A*1 was any one of these pixels from 20xx001th MOD09A1. Correspondingly in the same position, they were *A*2 pixel of 20xx009th MOD09A1 and the *B* pixel of 20xx001th MOD13Q1. The C pixel was the corresponding production date (The day of the year, DOY) of the *B* pixel, whose value ranged from 1 to 16 at the 20xx001th day.

Let us compare the results of the MOD09A1 and MOD13Q1 on the 20xx001th day. The following three steps exist in the overall process.
**First step**: If 1 ≤ *C* < 9: We consider *B* as accurate, and *D*1 is assigned to *B* if the case is the following: The result from MOD13Q1 (that is the 16-day optimal value with 250 m resolution) is more accurate than the one obtained from MOD09A1 (that is the eight-day optimal value with 500 m resolution).**Second step**: Ιf 9 ≤ *C* < 17 and *A*1 = *B*, this means that the two classification results are consistent, and *D*1 can be assigned either to *A*1 or *B*.**Third step:** If 9 ≤ *C* < 17 and *A*1 ≠ *B*, we need to compare their correlation with the surrounding pixels to determine the final result. A gradient equation is being proposed in this study (Equation (1)). Comparing the gradient values of *A*1 and *B*, if $gradient_{A1}<gradient_{B}$, the correlation between *A*1 and the surrounding pixels in the window can be considered more reasonable than *B*, so *D*1 = *A*1; otherwise *D*1 = *B*.
(1)$$gradient_{i}=\sqrt{\frac{\sum_{i,t=1\sim T}^{T}{\left(dy_{it}/dx_{it}\right)}^{2}}{T}}$$
where *T* is the number of surrounding pixels in the window, $dx_{it}$ is the distance of pixel *i* and the surrounding pixel *t*, and $dy_{it}$ is the difference between them. If the pixel window is set to 3 × 3, *T* = 8, $dx_{it}$ = 1 or $\sqrt{2}$, and $dy_{it}$ = 0 or ±1.

These steps can also be expressed as the follows:
(2)$$D1=\left\{ \begin{array}{ll} B & \left(1\leq C<9\right)\\ A1 & \left(9\leq C<17\&A1=B\right)\\ A1 & \left(9\leq C<17\&A1\neq B\&gradient_{A1}<gradient_{B}\right)\\ B & \left(9\leq C<17\&A1\neq B\&gradient_{A1}\geq gradient_{B}\right) \end{array}\right.$$
where *D*1 is the pixel of fusion results for the 20xx001th day.

According to the above process, all pixels of the 20xx001th day results can be calculated. Likewise, using *A*2, *B*, and *C*, all pixels *D*2 corresponding to the results of the 20xx009th-day can also be produced. The *D*2 can be expressed as follows:
(3)$$D2=\left\{ \begin{array}{ll} B & \left(9\leq C<17\right)\\ A2 & \left(1\leq C<9\&A2=B\right)\\ A2 & \left(1\leq C<9\&A2\neq B\&gradient_{A2}<gradient_{B}\right)\\ B & \left(1\leq C<9\&A2\neq B\&gradient_{A2}\geq gradient_{B}\right) \end{array}\right.$$

2. Comparison of the different sized windows

Regarding the HDSTAFM process, the first and the second steps depend only on the classification results and process most of the pixels. The third step is mainly performed on the pixels that are related to two inconsistent classification results with different periods. These results are affected by the window size. The primary reference window sizes are 3 × 3, 5 × 5, and 7 × 7. In order to determine the best window size, an illustrative example is shown in the following Figure 6. Pixels *A*11 and *A*21 are at the junction of water and land, where *A*11 = 0 and *A*21 = 1. The value of the fused pixel *A*1 is equal to 1 (*D*1 = *A*21) for 3 × 3 or 5 × 5 windows, whereas it is equal to 0 (*D*1 = *A*11) for the 7 × 7 window. This example illustrates that the water-and-land boundary pixel tends to the “water” result of MOD13Q1 when the window is small; otherwise, it tends to the “non-water” result of the MOD09A1 when exceeding a certain range. 

Pixels *A*12 and *A*22, are also at the junction of water and land, but *A*12 = 1 and *A*22 = 0. The resulting value of the fused pixel A2 is equal to 1 (*D*2 = *A*12) for the 3 × 3 window, and it is equal to 0 (*A*22 = 0) for the 5 × 5 or 7 × 7 windows. This example illustrates that when the window is small, the “water-and-land” boundary pixel tends to the “water” result of MOD09A1; otherwise, it tends to the “non-water” outcome of MOD09A1 when exceeding a certain range. Combining the two examples, the fusion results of *A*1 and *A*2 are both biased towards the “water” for a smaller window and towards the “non-water” for a larger window. Moreover, it indicates that 5 × 5 is the best alternative window size compared to 3 × 3 and 7 × 7. *A*13 and *A*23 pixels, are relatively far from a wide range of water bodies but *A*23 is recognized as “water” because of its higher resolution or we have a potential misidentification when the fusion results are biased towards “non-water”. 

In general, the 5 × 5 window is optimal compared to 3 × 3 and 7 × 7, although some separate “water” or “land” results from the MOD13Q1 may be eliminated.

### 3.4. Accuracy Assessment

Some cloudless Landsat images were used to test the results. This study has employed the NDWI to initially extract “water” bodies and to manually correct them. Overall accuracy (OA), user’s accuracy (UA), producer’s accuracy (PA), and kappa coefficients (KC) have been calculated.

## 4. Results

### 4.1. Comparison of the Results and Accuracy Assessment

#### 4.1.1. Comparison of the Results of Different Windows

Figure 7b–d shows that the fusion results of the 3 × 3, 5 × 5, and 7 × 7 windows in Poyang Lake have all been feasible for the methodology. However, some details also exist for small water bodies, especially for complex ones, which are characterized by algae, clouds, and their spatial resolution. The “non-water” part of the 3 × 3 window result is relatively large, and some “water” pixels and plaques have not been extracted. The result of the 7 × 7 window is relatively small, and some “non-water” pixels have been treated as “water”.

Similarly, Figure 8b–d shows the fusion results of 3 × 3, 5 × 5, and 7 × 7 windows in Hulun Lake, respectively. There are both obvious misclassification in “water” and “non-water” plaques in the result of the 3 × 3 window (Figure 8b). Most of the “water” pixels have been extracted but there are also some subtle “water” pixels or plaques that have been misclassified in the result of the 7 × 7 window (Figure 8c). Compared to the result of the 7 × 7 window, the advantage of the 5 × 5 window is mainly reflected in the better extraction effect on some fine “water” bodies.

#### 4.1.2. Accuracy Assessment with Landsat Images

Some classification results generated by the methodology have been validated by comparing with some cloudless Landsat-based results in the same period (Table 3). The OA, UA, PA, and KC of the four cases with eight images are almost all high. There are some shortcomings for Poyang Lake, where many narrow “water” bodies are not easily detected by lower spatial-resolution MODIS data.

### 4.2. Case Monitoring Results

The results of Bosten, Namco, Hulun, and Poyang lakes for the period 2000–2016 have been acquired.

#### 4.2.1. The Case of Bosten Lake

Water extent variations are mainly located in the north and west of Bosten Lake (Figure 9a). The maximum “water” area occurred in 2003/5/1 (Figure 9b) and the minimum in 2012/1/1 (Figure 9c). The “water” area is consistent with the relative height obtained by the USDA, DAHITI, and LEGOS, respectively (Figure 9d). The number of valid auxiliary data can be seen in Table 2, and the correlation coefficients between “water” area and relative height, are 0.76 (USDA), 0.87 (DAHITI), and 0.88 (LEGOS). The quality of the USDA data is relatively low (Figure 9e).

#### 4.2.2. The Case of Namco Lake

The “water” area of Namco Lake has shown an increasing trend, but it is minimal compared to the entire lake (Figure 10a). The maximum water area occurred in 2016/6/2 (Figure 10b) and the minimum in 2000/7/2 (Figure 10c). The increase of the lake area mainly occurred in 2000–2009, and it is relatively stable, but a slight increase trend occurred in the period 2010–2016 (Figure 10d). The results have proven to be reasonable, where the correlation coefficient was found to be *R* = 0.79 in the case of “water” area and “relative height” for the LEGOS.

#### 4.2.3. The Case of Hulun Lake

The water extent of Hulun Lake has presented large variations in the period 2000–2016, which can be divided into two phases: It has decreased in the period 2000–2010 and increased in the time interval 2011–2016 (Figure 11d). The maximum “water” area occurred in 2003/4/7 (Figure 11b), and the minimum in 2010/11/9 (Figure 11c). The “water” variations mainly occurred in the northeast and southeast of the lake. Figure 11e presents the relationship between the “water” area and the “relative height” in Hulun Lake. They are equal to 0.93, 0.93, and 0.92 for the USDA, DAHITI, and LEGOS, respectively, which indicates the fact that the obtained results are quite reliable (Table 2).

#### 4.2.4. The Case of Poyang Lake

There are obvious dynamic variations in the water extent for Poyang Lake. The maximum “water” area was as high as 3413.51 km^2^ (Figure 12b) whereas the minimum was 886.03 km^2^ (Figure 12c) and they were recorded on 2016/7/20 and 2004/1/25, respectively. 

Seasonal dynamic variation of water extent in Poyang Lake is prominent and regular (Figure 12d). The correlation coefficients between the “water” area and the “relative height” are 0.83 (DAHITI) and 0.46 (LEGOS) (Figure 12e). 

It looks like the “water” extraction results are poor compared with the “relative height” from LEGOS. Therefore, we further compared the relationship between the “water” area and the observed “water” level from the Hukou Station. The results have shown that this correlation is good with *R* = 0.90 (Figure 13b). It shows that the “water” extraction result is reasonable, and that the LEGOS data may have some defects.

## 5. Discussion

### 5.1. Advantages of the Approach

The RF classifier, a robust and efficient machine learning method, can perform well in the case of this preliminary classification of “water” and “land”. However, noise such as clouds is inevitable for the time series of “water” extraction. In order to ensure the reliability and continuity of the results, post-processing including de-noise and “water” restoration has been built based on existing research in this paper. The denoising process mainly used QA layer data and slope data. The “water” restoration included two steps. In past research, the first step commonly used technical means by considering the results of adjacent images. However, many “water” pixels were still not restored, so the second step aimed to further improve the results by using the elevation data. It has to be emphasized that this process mainly relies on the accuracy of the DEM data, where the current ASTER GDEM data basically meet the requirements, but they still need to be improved in the future.

The HDSTAFM provides a good approach for data fusion in remote sensing applications. It has two main advantages: (1) It uses multiple sets of data from the same data source (MODIS), and it takes into account the temporal and spatial information for each pixel; and (2) it performs fusion only for the classification results, which reduces errors caused by the fusion method itself. In fact, compared to the classification results of MOD09A1 or MOD13Q1, the accuracy of the fusion result is higher [18]. The HDSTAFM utilizes more bands information and reduces the impact of noise because of the integration of the two results. In addition, it must be mentioned that the window size is set to 5 × 5, which looks better than 3 × 3 and 7 × 7. The two examples (A1 and A2) have shown that the fusion results of the 5 × 5 window are more robust than the other two, and the example A3 has clearly proven that the fusion results can eliminate some small separate “water” or “land” plaques that may cause noise (Figure 6).

### 5.2. Challenges for the Four Lakes

Although the classification results have proven to be reliable, several challenges are still existing in mapping time series “water”, such as clouds, snow or ice, terrain or cloud shadows, and turbidity. The noise varies significantly for these four lakes due to their different “water” types, climates, and underlying surfaces. 

Namco Lake is the only saltwater lake among them. Its “water” type is relatively simple with little sediment or algae, and there are few other features around it that can interfere with the “water” classification. The results are mainly affected by snow, ice, and clouds. The “water” type of Bosten Lake is also relatively simple, but there are some vegetation and artificial buildings as well as the “water” is relatively shallow on the west side of the lake. Other noises such as clouds, cloud shadows, snow and ice are relatively few. The classification results of the lake are good despite some noise effects. Hulun Lake is less affected by the cloud. However, the salinity of the “water” is higher than in the freshest “water” cases. Due to the large fluctuation of the “water” extent and temperature, the classification results have shown some differences. 

These problems from the three lakes have been well resolved by the approach, according to the test results (Table 3). The classification results of Poyang Lake are relatively poor. Many types of noise have a severe effect of the “water” extraction, including clouds, clouds shadows, turbidity, vegetation, and algae. Clouds and cloud shadows are common in Poyang Lake, especially in spring and summer. There are no completely reliable methods to remove their effect; however, some serious attempts have been made in this paper (Section 3.2). In addition to the noise, the low spatial resolution of the MODIS data is the main influencing factor for accuracy. There are many small “water” patches and narrow river channels in the Poyang Lake region. They are challenging to be detected using MODIS products, but they can be well detected using Landsat images. Overall, the results of Poyang Lake are reliable (Table 3 and Figure 12 and Figure 13).

### 5.3. Further Improvement

#### 5.3.1. Potential Improvement to Remove Noise

Some improvements can still be worth considering, including denoisng and “water” restoration. Zhu et al. [12] detected clouds, cloud shadows and snow for Landsat 4–8 by improving the fmask algorithm. The multi-satellite methodology helps to avoid the noise [31,32]. In addition to optical remote sensing data, radar and microwave remote sensing data are partially suitable for “water” extraction [33,34]. Some optical remote sensing images with much cloudiness can be replaced by radar data during the same period.

#### 5.3.2. Selection of More Efficient Classification Methods

The RF classifier is more efficient and robust than some traditional methods. However, it will also appear obvious misclassification when the samples are poorly representative. Some new classification methods proposed have also been applied successfully towards “water” classification [35,36]. Unfortunately, there are no classification methods that can classify “water” and “land” without being influenced by a certain level of noise so far. Robust and efficient classifiers are still our main pursued goal. 

#### 5.3.3. Applications of the Fusion Method for Higher-Resolution Data

The HDSTAFM mainly considers the temporal and spatial relationships of each pixel, where the temporal and spatial features are compared through a gradient value for the adjacent time and the neighboring pixels. The fusion results have proven to be more robust and accurate, and a higher spatiotemporal resolution has been achieved. This method can also be considered to fuse other data sources, such as Landsat time series data. Even different data can still refer to the fusion idea if the temporal and spatial relationship is satisfied.

## 6. Conclusions

This study aimed to map continuous lake water extents with high spatiotemporal resolution. It is conducive to more accurately understand the “water” variations of the lakes. Four different types of lakes have been selected and used as application cases. The approach comprises of three basic parts: (1) Preliminary water classification using the RF classifier; (2) post-processing, including de-noise and water restoration; and (3) water result fusion using the HDSTAFM method. Continuous water results at high frequency (every eight days) and moderate spatial resolution (250 m) have been provided for these four lakes.

Some specific conclusions can be conducted:
The classification results are reliably validated using some Landsat-based results. The approach performs efficiently in obtaining time series “water” extent of the lakes based on the MODIS products by comparing against altimetry data and water level data.The RF classifier is efficient in the long-term sequence classification, and post-processing is also critical in removing noise and restoring water bodies.Although there are still some shortcoming (e.g., single water pixel information may be eliminated), the HDSTAFM is feasible for the fusion of two sets of water results. Moreover, the 5 × 5 window performs better than the 3 × 3 and 7 × 7 windows.

In general, the approach is effective for continuous long-term monitoring of the lake “water” extent. In the future, we will try to improve it further and mapping dynamic “terrestrial water”, including several lakes and rivers. 

## Figures and Tables

**Figure 1 sensors-19-04873-f001:**
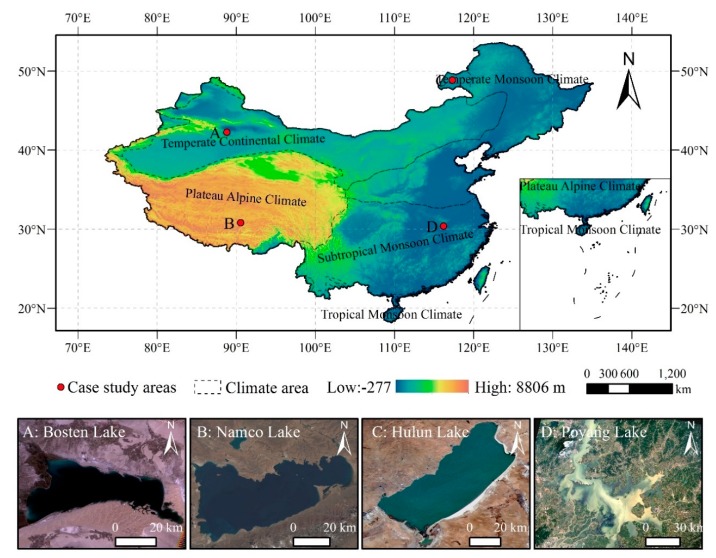
Location of four case study areas. Landsat true color images show the water body extent of the case study areas.

**Figure 2 sensors-19-04873-f002:**
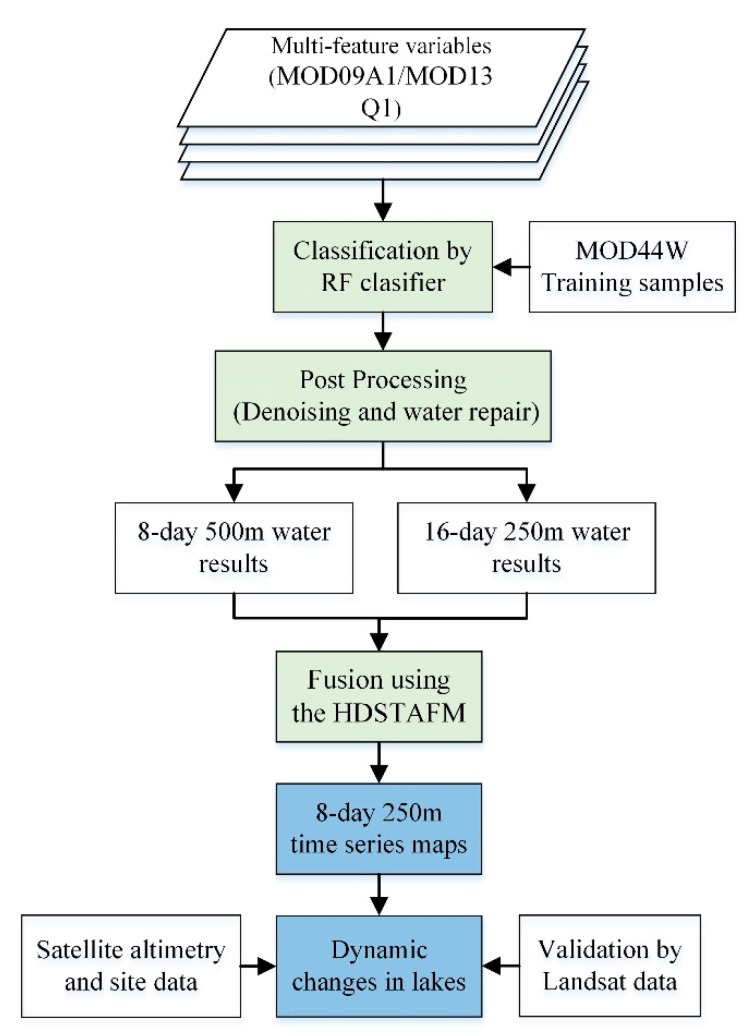
The flowchart of the research methodology.

**Figure 3 sensors-19-04873-f003:**
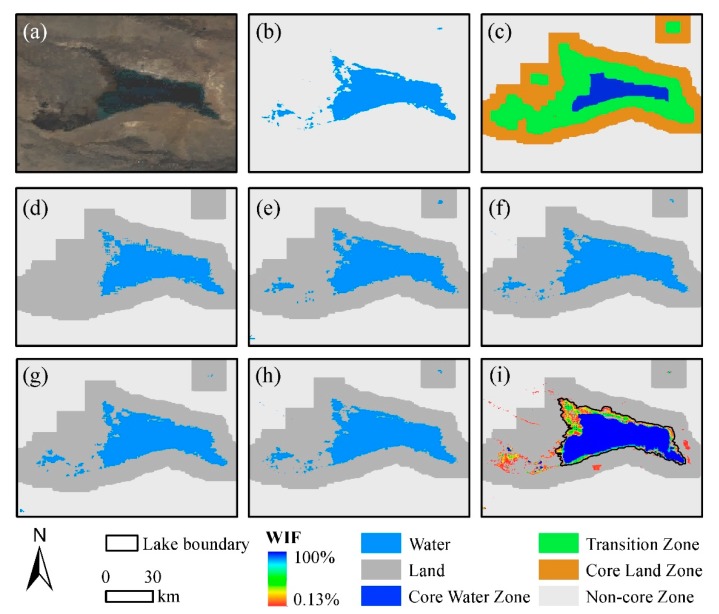
The example of automated extraction of water extent and water fusion result in Bosten Lake. (**a**) False-color composite (bands 1, 4, and 3) on 2,016,081th day; (**b**) the MOD44W water result; (**c**) the buffer, including CWZ, TZ, and CLZ; (**d**,**e**) water results of 2,016,081th and 2,016,089th days from MOD09A1, respectively; (**f**) the water result of 2,016,081th day from MOD13Q1; (**g**,**h**) fused water results of 2.016,081th and 2,016,089th days, respectively; and (**i**) the water inundation frequency (WIF) map.

**Figure 4 sensors-19-04873-f004:**
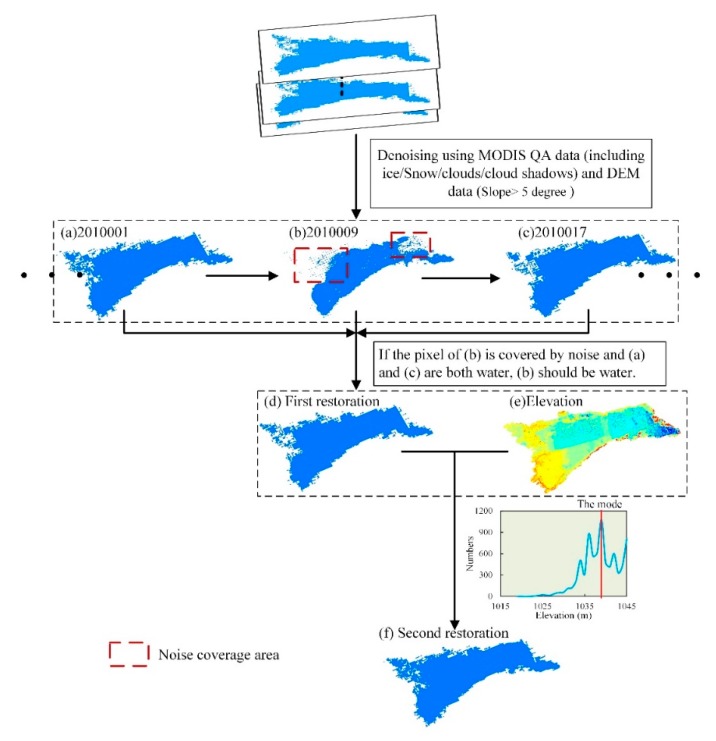
An illustrative example showing the water restoration after de-noise. (**a**–**c**) The denoised water results in Bosten Lake on 2,010,001th and 2,010,017th days, respectively; (**d**,**f**) the first and second water restoration results; and (**e**) the elevation data.

**Figure 5 sensors-19-04873-f005:**
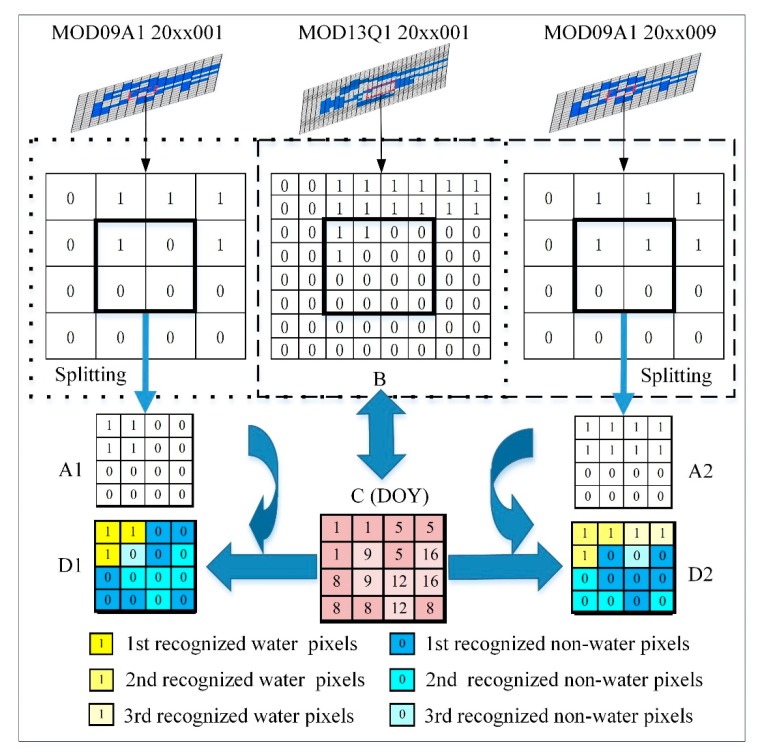
Fusion schematic of two sets of water results within 500 m every eight days and 250 m every 16 days for the window 3 × 3 (source: Rao et al. [18]).

**Figure 6 sensors-19-04873-f006:**
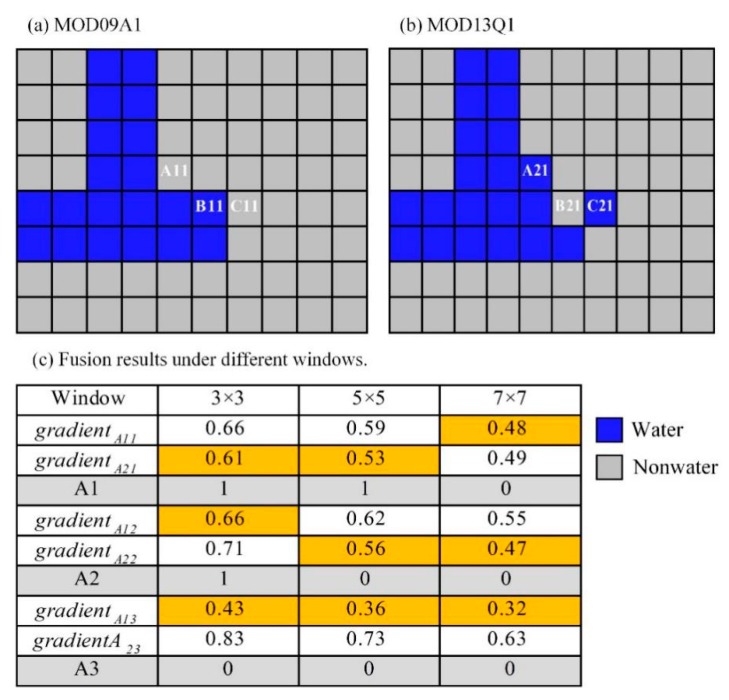
An illustrative example showing the fusion results under different windows when both the period and the value of the two results are inconsistent. (**a**) The schematic of MOD09A1; (**b**) the schematic of MOD13Q1; and (**c**) fusion results under different windows.

**Figure 7 sensors-19-04873-f007:**
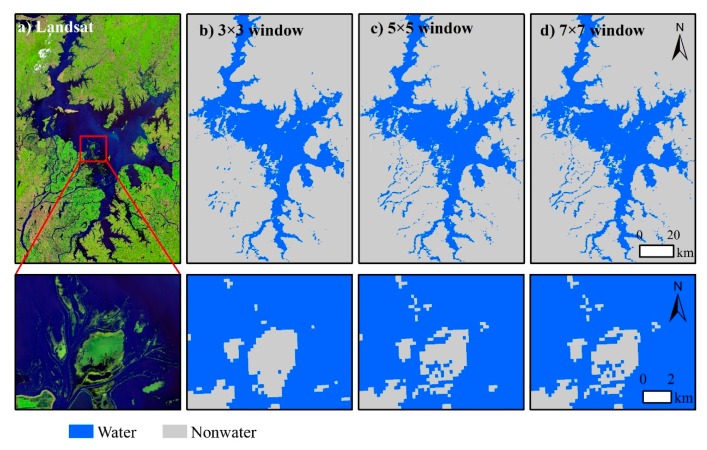
An illustrative example to compare water results for different windows in Poyang Lake. (**a**) False color composite on 23 June 2016. Fusion results on 2016169th day with (**b**) 3 × 3, (**c**) 5 × 5, and (**d**) 7 × 7 windows.

**Figure 8 sensors-19-04873-f008:**
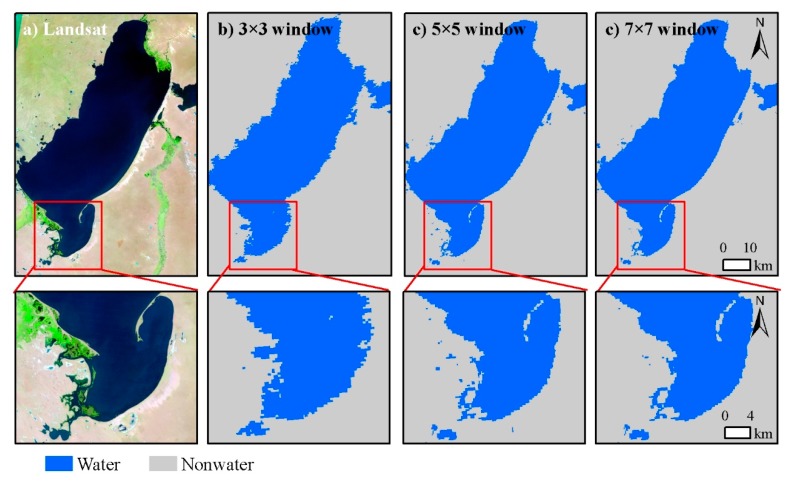
An illustrative example to compare water results for different windows in Hulun Lake. (**a**) False color composite on 28 October 2011. Fusion results on 2,011,297th day with (**b**) 3 × 3, (**c**) 5 × 5, and (**d**) 7 × 7 windows.

**Figure 9 sensors-19-04873-f009:**
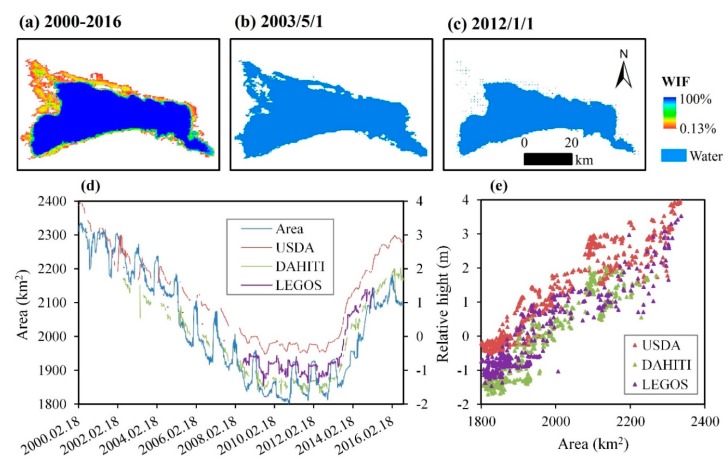
Spatial distribution and temporal variation of Bosten Lake. (**a**) The WIF map in 2000–2016; (**b**) water body map of 2003/5/1, with the largest water area in 2000–2016; (**c**) water body map of 2012/1/1, with the smallest water area in 2000–2016; (**d**) time series of the water area (km^2^) and relative height (m) in 2000–2016; and (**e**) correlation between water area (km^2^) and relative height (m).

**Figure 10 sensors-19-04873-f010:**
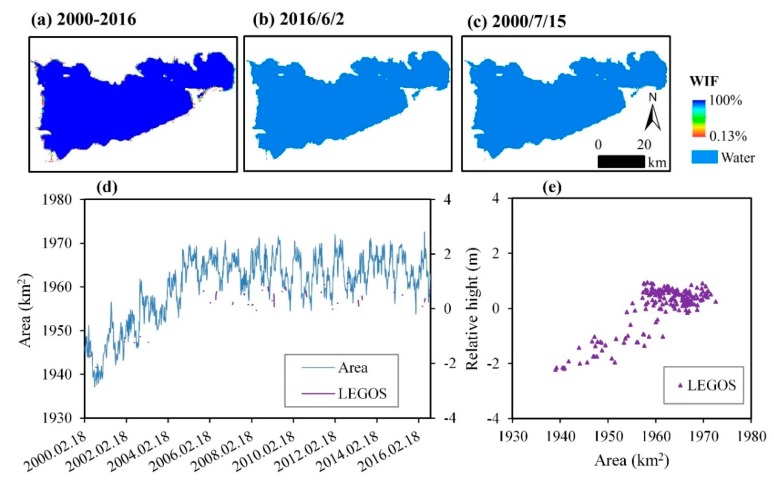
Spatial distribution and temporal variation of Namco Lake. (**a**) The WIF map in 2000–2016; (**b**) water body map of 2016/6/2, with the largest water area in 2000–2016; (**c**) water body map of 2000/7/15, with the smallest water area in 2000–2016; (**d**) time series of the water area (km^2^) and relative height (m) in 2000–2016; and (**e**) correlation between water area (km^2^) and relative height (m).

**Figure 11 sensors-19-04873-f011:**
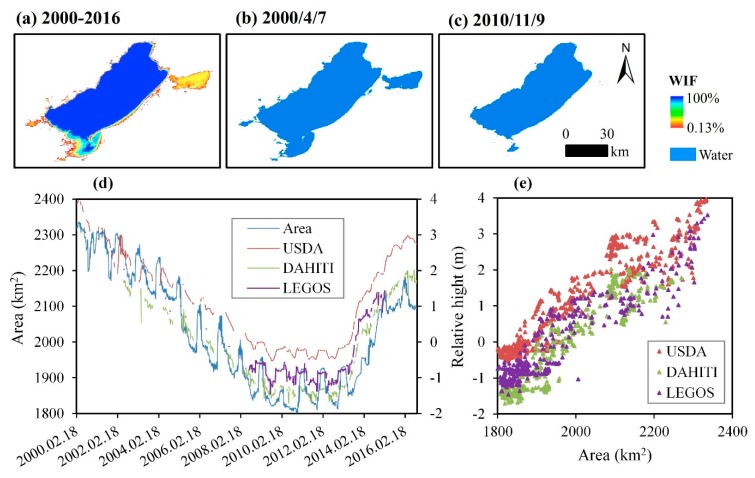
Spatial distribution and temporal variation of Hulun Lake. (**a**) The WIF map in 2000–2016; (**b**) water body map of 2000/4/7, with the largest water area in 2000–2016; (**c**) water body map of 2010/11/9, with the smallest water area in 2000–2016; (**d**) time series of the water area (km^2^) and relative height (m) in 2000–2016; and (**e**) correlation between water area (km^2^) and relative height (m).

**Figure 12 sensors-19-04873-f012:**
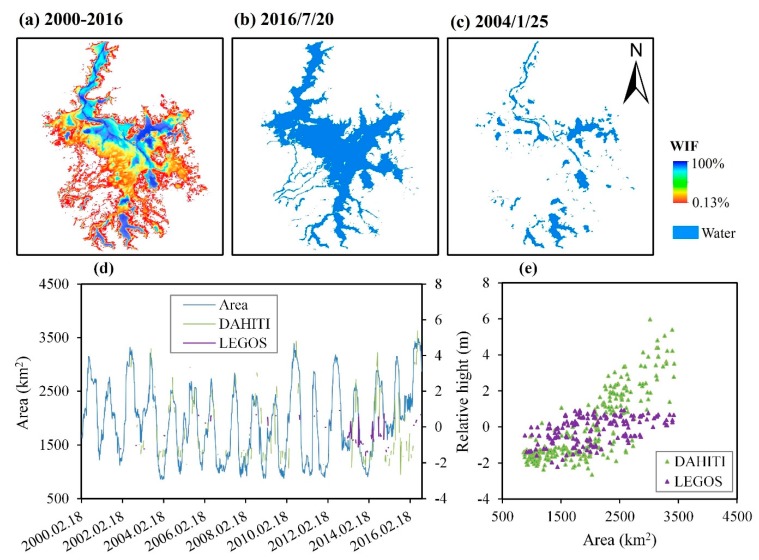
Spatial distribution and temporal variation of Poyang Lake. (**a**) The WIF map in 2000–2016; (**b**) water body map of 2016/7/20, with the largest water area in 2000–2016; (**c**) water body map of 2004/1/25, with the smallest water area in 2000–2016; (**d**) time series of the water area (km^2^) and relative height (m) in 2000–2016; and (**e**) correlation between water area (km^2^) and relative height (m).

**Figure 13 sensors-19-04873-f013:**
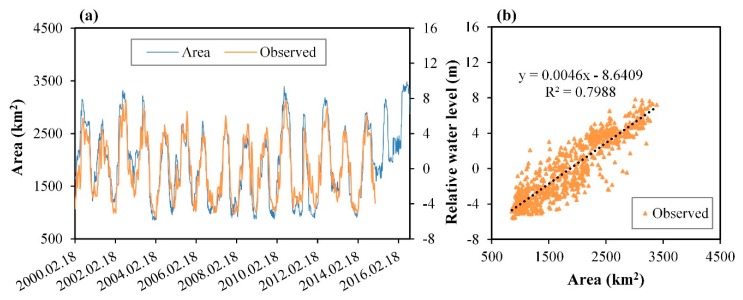
The relationship between the water area and water level. (**a**) Time series of the water area and observed relative water level in 2000–2016. (**b**) The correlation between them.

**Table 1 sensors-19-04873-t001:** Basic details of the four lakes.

Lakes	Location	Elevation (m)	Area (km^2^)	Maximum Depth (m)	Average Depth (m)	Salinity (g · L^−1^)	Climate	Annual Precipitation (mm)
Bosten	E86°40’–87°25’ N41°56’–42°14’	1048	992	16.5	8.08	1.865	Temperate Continent	121
Namco	E88°33’–89°21’ N31°34’–31°51’	4718	1920	>95	–	1.78	Plateau Alpine	434
Hulun	E117°00’–117°42’ N48°30’–49°21’	545	2339	8	5.92	1.17	Temperate Continent	244
Poyang	E115°47’–116°45’ N28°22’–29°45’	10	3283	25.1	8.4	0.047	Subtropical Monsoon	1500

**Table 2 sensors-19-04873-t002:** The number of altimetry data obtained and correlation coefficients corresponding to water areas.

Lakes	The Number of Valid Data			Linear Correlation (R)		
	USDA	DAHITI	LEGOS	USDA	DAHITI	LEGOS
Bosten	476 (2000–2016)	384 (2002–2016)	252 (2002–2015)	0.76	0.87	0.88
Namco	–	–	169 (2000–2016)	–	–	0.79
Hulun	543 (2000–2016)	389 (2000–2016)	371 (2000–2015)	0.93	0.93	0.92
Poyang	–	261 (2002–2016)	156 (2000–2016)	–	0.82	0.46

“–”: No data available.

**Table 3 sensors-19-04873-t003:** Validation of MODIS-based classification results using Landsat-based reference results.

Lakes	Date	OA	UA	PA	KC
Bosten	16/July/2003 (High)	0.95	0.93	0.94	0.90
	24/September/2011 (Low)	0.97	0.99	0.92	0.93
Hulun	16/June/2000 (High)	0.97	0.99	0.97	0.94
	28/October/2011 (Low)	0.97	0.96	0.98	0.95
Namco	02/October/2015 (High)	0.97	0.99	0.96	0.93
	18/November/2003 (Low)	0.97	1.00	0.96	0.93
Poyang	16/February/2016 (High)	0.94	0.92	0.91	0.86
	23/June/2016 (Low)	0.91	0.85	0.79	0.78

NOTE: For each of the four lakes, dates with high and low water extent were selected.

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
