# Peer review of "Continuous Dynamics Monitoring of Multi-Lake Water Extent Using a Spatial and Temporal Adaptive Fusion Method Based on Two Sets of MODIS Products"

_sensors, 2019, doi:10.3390/s19224873_

Round 1

Reviewer 1 Report

Line 16-30: I do not see any significant progress in already existing research. I understand from the abstract that the authors propose a methodology to create a spatiotemporally continuous MODIS based satellite dataset for detecting water bodies. But that can be done by using different datasets from MODIS itself.

Line 19: “diverse regions of China”? The authors need to be specific here. Whether these lakes are located over different climate regions and altitude?

Line 22: I am a bit confused here. Why not using the MOD10A1 or MOD10A2 which has already inland water bodies specified as pixel value 237.

Line 91: Aster is also Remote Sensing data and this section must be integrated with “Remote Sensing Datasets”. Or maybe change the “Remote Sensing Datasets” section to MODIS datasets because in the next section the authors talk about Landsat datasets which are again a remote sensing dataset.

Line 77-95: The authors must give the use of the data in the present study. Mere enlisting the data and where to download them with some specifications is not enough.

Line 98: The research problem which the authors are trying to solve is removing the noise from cloud and cloud shadows, is with The Landsat images as well. Actually the low temporal resolution (16 days) is a big problem with Landsat which the authors are trying to use for validation.

I will stop reviewing the manuscript here and enlist the problems which I see in the conclusions drawn from the manuscript.

The classified inland water pixels are available with pixel value 237 in MOD10A1 and MOD10A2. If one wants to have an accurate estimation of the number of water pixels over a period of time with monthly frequency, they can easily take daily data into consideration and take the average of the data without the cloud cover. I do not see any use of using a classifier and making the process more complicated if the aim of the study is to make a continuous time-series of lake water area over a period of time. Also, the cloud is a problem with Landsat data too. The aim of the study is confusing. I do not understand what the authors have been trying to achieve by post-processing and classifying the data when the inland water is already available with MODIS.

Author Response

Dear reviewer:

         Thank you very much for your careful reading of the manuscript and some comments and suggestions. I have made some modifications and specific explanations for the manuscript given to your comments and suggestions. 

          Looking forward to your more valuable suggestions, Sincerely!

Reviewer 2 Report

The manuscript entitled " Continuous dynamics monitoring of multi-lake water extent at high spatiotemporal resolution using MODIS data" by Pinzeng Rao and co-authors.

They developed a new framework of monitoring changes in water bodies using satellite images.

Overall, the article is written well, and I do not have any critical comment. However, the introduction is very slim which could be extended. The discussion should be extended to relate the study with a global perspective and implicating of the study elsewhere in the world

Author Response

(The authors gave the same response as above.)

Reviewer 3 Report

Rao et al. presents the lake mapping using MODIS data across China. This study is interesting and suitable to publish in Sensor. Some comments below are suggested to improve this manuscript before potential publication.

Title: “high spatiotemporal resolution”, MODIS has a sparse spatial resolution. Please correct your title.

L34-37: These short sentences could be combined into two long sentences. Also, suggested citation here for a new Chinese lake study such as http://dx.doi.org/10.1016/j.rse.2018.11.038

L60: Namco Lake-> Nam Co, and elsewhere

500 m MOD09A1 and 250 m MOD13Q1 are used. Two data sets have different temporal and spatial resolutions. How do you fuse them? How about the uncertainty for time series of water mapping?  

Figure 1: legend, Low -155 and High 8460 m? Please correct it.

“2.4. Validation and auxiliary data” The validation data for lake area should have the same date with MODIS data. However, the data downloaded here is old. How to match them? A suggested website for China lake dataset at https://data.tpdc.ac.cn.

Location of Figures should match with their texts.

Author Response

(The authors gave the same response as above.)
